# Self-Stigma and Mental Health in Divorced Single-Parent Women: Mediating Effect of Self-Esteem

**DOI:** 10.3390/bs13090744

**Published:** 2023-09-06

**Authors:** Anna Kim, Sesong Jeon, Jina Song

**Affiliations:** 1Ulsan Public Agency for Welfare Family Promotion Social Service, Ulsan 44717, Republic of Korea; skyplane1022@uwfdi.re.kr; 2Major in Child & Family Studies, School of Child Studies, College of Human Ecology, Kyungpook National University, Daegu 41566, Republic of Korea; bgnyoto@knu.ac.kr

**Keywords:** mental health, self-stigma, self-esteem, divorced single-parent women, mediating effect

## Abstract

Numerous studies have addressed the issue of “self-stigma” among divorced single-parent women. However, there is a scarcity of quantitative data available on this subject. Moreover, while self-esteem is a crucial factor throughout life, it has been extensively studied in the context of “children” from single-parent families, but not from the perspective of parents themselves. To address this gap, the present study aimed to explore the relationship between self-stigma, self-esteem, and mental health in 347 divorced, single-parent women. The online survey recruited participants randomly, with a specific focus on single mothers who were divorced and had more than one child under the age of 18. The analysis involved utilizing SPSS 25.0 (IBM Co., Armonk, NY, USA) and PROCESS Macro Version 4.1 (Model 4) to conduct descriptive statistics, frequency analysis, reliability assessment, correlation analysis, and mediating analysis. The findings revealed that self-esteem played a partial mediating role in the relationship between self-stigma and mental health. In other words, higher levels of self-stigma among divorced, single-parent women were associated with poorer mental health outcomes. Additionally, the study discovered that engaging in more self-stigma was linked to lower self-esteem and increased mental health distress. These results underscore the significance of internal factors, such as self-stigma and self-esteem, and highlight their relevance in formulating policies aimed at supporting divorced single-parent women. Policymakers should take these factors into account to develop effective strategies to aid this specific group.

## 1. Introduction

Despite being a prevalent family structure in contemporary society, divorced single-parent women have been extensively studied as a type of vulnerable family, facing complex challenges [1,2,3,4,5]. However, psychological difficulties persist without a clear resolution [6,7,8,9,10,11]. These women encounter various issues, such as depression, anxiety, sleep disorders, emotional and physical fatigue, and excessive anxiety [6]. Moreover, they face more obstacles and struggles in terms of economic conditions and employment compared to single-parent men [10,12,13,14]. Additionally, single-parent women not only experience lower levels of happiness compared to non-single-parent women but also report negative mental health conditions, such as isolation, anxiety, depression, paranoia, and even suicidal thoughts [11,15]. Due to these concerning findings, it remains crucial to continue investigating the mental health challenges that divorced single-parent women confront and find ways to support them.

### 1.1. Self-Stigma and Mental Health

Stigma can be categorized into two main types: social stigma and self-stigma. These distinctions are based on who holds negative judgments about the stigmatized group. “Social stigma” occurs when the general public harbors prejudices against the stigmatized group, whereas “self-stigma” refers to the internalization of public stereotypes by individuals about themselves and their specific situations [16]. Corrigan and his colleagues proposed a process model named *Awareness*, *Agreement*, *Application*, *and Harm*, which outlines the progression of self-stigmatization and its harmful consequences. This process involves individual becoming aware of public stigma (Awareness), agreeing with the negative beliefs directed at their group (Agreement), and applying those negative beliefs to themselves (Application), resulting in harmful outcomes, such as reduced self-esteem (Harm) [17,18]. In other words, social stigma precedes self-stigma, implying that divorced single-parent women exist in a society where divorce and stigmatization of single-parent families are prevalent. Consequently, they are more likely to suffer from a negative reputation due to the widespread social stigma.

According to single-parent women themselves, their mental health struggles and psychological challenges are often seen as a natural consequence of their social circumstances. This belief negatively impacts their willingness to seek help, accept support, or engage in psychological interventions [11]. Despite some reduction in negative attitudes toward divorce, the stigma surrounding divorced single-parent women persists in a society that primarily upholds the norm of nuclear and two-parent families [19]. Moreover, the prevailing labor market and social welfare systems place single-parent women in subordinate and economically disadvantaged positions, leading them to perceive judgment from others [3]. A study conducted by Zartler [19] revealed that parents and children from two-parent families tend to view single-parent families as deficient, which further contributes to feelings of inadequacy among parents and children in single-parent households. This situation is particularly challenging for single-parent women, who have limited personal resources and, therefore, fewer opportunities to receive support for fostering a positive self-identity. Consequently, they are more likely to internalize public or professional judgments, including those propagated by the media [9].

Self-stigma has detrimental effects on the well-being of divorced, single-parent women, making it difficult for them to maintain and improve their vulnerable mental health. Internalizing stigma leads to feelings of failure, shame, and guilt among divorced women [20], while single-parent women experience self-destructive shame due to societal stigma [8]. These recurring emotions pose a risk of developing negative mental health outcomes. Moreover, the self-stigma experienced by single-parent women reinforces existing stereotypes, causing them to perceive their current distressing mental health status as justifiable [11]. Individuals who engage in self-stigmatization and are aware of discrimination are more likely to suffer from poor mental health, and this pattern applies to divorced, single-parent women who are prone to self-stigma.

### 1.2. Self-Stigma and Self-Esteem

Self-esteem refers to an individual’s positive or negative attitude towards themselves [21]. Life events such as divorce and single parenthood can significantly impact a person’s self-esteem. Previous studies focusing on parents from single-parent or divorced families consistently found that they tend to have lower self-esteem. A study conducted by Bleidorn and colleagues [22] revealed varying results for each individual, but on average, divorcees experienced a significant decline in self-esteem before and after the divorce or separation. Comparisons between divorced individuals, single-parent women, and married individuals indicated that divorced and single-parent women generally exhibited lower self-esteem than their married counterparts [23,24]. Furthermore, research investigating the association between partner presence and self-esteem demonstrated that parents in single-parent households had lower self-esteem than parents in married or cohabitating families, with this difference being more pronounced for women than for men [25]. Moreover, single-parent women face challenges in gaining personal and social respect as they navigate the balance between paid labor and prioritizing parenting responsibilities [8]. Particularly, those with limited support networks often experience lower self-esteem, economic difficulties, and relationship conflicts due to the complexity and severity of their burdens [26].

Self-stigma can have adverse effects on the self-esteem of divorced, single-parent women. By internalizing this self-stigma, they may feel inferior as parents [27], leading to a diminished sense of self-worth and weakened belief in their ability to achieve their goals [17,28]. The societal stigma surrounding single-parent women is closely linked to their personal resources, such as self-esteem and self-confidence. Particularly, single-parent women in vulnerable situations, who struggle with psychological difficulties, are more likely to internalize negative judgment from the public and perceive themselves as “failed mothers”. This self-criticism contributes to lower self-esteem. On the other hand, mothers who do not internalize the public stigma challenge stereotypes and demonstrate more positive beliefs about themselves [9]. In conclusion, frequent experiences of shame and guilt resulting from the internalized stigma can lead divorced women to worry about how others perceive them. This can lead to self-doubt, decreased confidence, and feelings of inadequacy, affecting their ability to pursue future job opportunities or form new relationships [20].

### 1.3. Self-Esteem and Mental Health

The consistent relationship between self-esteem and unfavorable feelings (i.e., depression, anxiety) shows a negative correlation. Sowislo and Orth [29] conducted a meta-analysis, revealing that low self-esteem is a significant factor in the development of negative mental health conditions like depression and anxiety. Numerous studies have emphasized that self-esteem is a stable and influential psychological factor [30], and Corrigan and colleagues [28] found that poor self-esteem adversely affects depression and resiliency. Especially, Peden and associates [5] found a significant correlation between self-esteem and depression in low-income single-parent women, suggesting that self-esteem influences depression. Similarly, Hatcher and colleagues [31] observed negative correlation between self-esteem and depressive symptoms in women from low-income single-parent families, where negative thoughts played a mediating role. In the context of black women from single-parent households, Atkins [32] also discovered a significant negative correlation between self-esteem, depression, and hostility.

In contrast, Taylor and Conger [33] provided insights into how certain inherent qualities of single mothers, such as self-esteem, self-efficacy, and positive coping strategies, play a role in enhancing their well-being and promoting positive mental health. Similarly, Lipman and colleagues [34] conducted interventions targeting self-esteem to foster well-being in women from single-parent families. Consequently, it is expected that self-esteem may have a positive impact on the vulnerable mental health status of women from divorced single-parent families. However, the existing research supporting this connection in the context of divorced single-parent families is limited, underscoring the need for further investigation and verification.

### 1.4. Present Study

This study aims to examine self-stigma and self-esteem as variables associated with the mental health of single-parent women. The health of Korean single parents is most significantly affected by societal prejudice, as evidenced by research conducted on divorced women raising children alone in East Asia [35]. Zhang’s research [36] revealed that single-parent women in Thailand experienced internalized stigma linked to divorce and unmarried pregnancy. Similarly, in China, due to societal stereotypes, the culture of emphasizing family unity, and prevailing maternity norms, women from divorced single-parent households faced adverse economic conditions, employment difficulties, strained relationships, and compromised physical and mental well-being [37]. Furthermore, single-parent women in Sri Lanka encountered similar challenges and were subjected to social stereotypes [38]. Single-parent women experienced greater social exclusion in many facets of their lives, which led to problems like depression [39].

In the absence of a partner, Lee [40] elucidates that the prevailing ideology surrounding gender roles can lead to detrimental emotional states. Within Korean society, the prevailing perception of “normal” and “healthy” families acts as a localized mechanism for discriminating against single parents. These parents are compelled to provide evidence of their financial struggles and requirements in order to receive support, thus subjecting them to discrimination and self-stigmatization [41].

Meanwhile, attitudes towards divorce have shifted, considering it as an opportunity to alleviate the burden of an unhappy marriage [22,42,43]. Additionally, certain single-parent women exhibit resistance against the challenges they encounter while maintaining a positive self-belief [8,9,26]. Given this perspective, it becomes crucial to investigate the progression of negative mental health conditions among women from divorced single-parent families. By examining the self-esteem of single parents in relation to the conventional understanding of gender roles in Korea, the study highlights the potential for negative effects. Accordingly, in this study, the following research problems were specifically identified.

Research question. Does self-esteem mediate the effect of self-stigma on the mental health of divorced single-parent women?

## 2. Materials and Methods

### 2.1. Participants and Procedure

This study focused on single parents residing in Ulsan, Republic of Korea, who required immediate policy attention between 30 July and 25 August 2021. The online survey targeting single parents residing in Ulsan was facilitated in collaboration with Ulsan Metropolitan City. Prior to participants engaging with the survey, a comprehensive explanation of its purpose was provided, and informed consent was obtained. Additionally, certain incentives were offered to participants as a token of appreciation for their response. Through online surveys with random sampling, participants were selected based on the reasons for their single parenthood, including divorce, spousal death, unmarried status, runaway partner, or spouse’s absence due to life-or-death circumstances. Among the 500 single-parents sampled, this analysis targeted only 347 divorced single-parent women. They met the following criteria: (1) raising more than one child under the age of 18; and (2) specifically, women who became single parents as a result of divorce. Table 1 displays the demographic information of the participants. More than half of the subjects were in their 40 s (58.2%), had graduated from high school (52.2%), and had an average monthly income ranging from more than 1 million won to less than 2 million won (59.9%). The most prevalent cases were single parenthood lasting for more than 5 years to less than 10 years (33.1%), followed by cases with a duration of less than 3 years (29.4%). The majority of women were employed (71.5%), with an equal proportion of women who had never received child support from their ex-spouse (50.7%) and those who had received child support more than once (49.3%). Many women reported having more than one social support network (65.7%), and the majority perceived their physical health as below-average (85.6%).

### 2.2. Measures

#### 2.2.1. Mental Health

In this research, the mental health of divorced single-parent women was assessed using the “mental health scale” developed by Goldberg and Hillier [44]. We used the Korean version of the mental health scale [45], which comprises 12 questions. Each question is rated on a four-point Likert scale, ranging from “never” (one point) to “always” (four points). To account for positive questions (1, 3, 4, 7, and 12), we applied reverse scoring, where a higher score indicated a greater level of mental health distress. Specific examples of the questions include inquiries about “feeling unhappy or depressed”, “always feeling anxious”, and “feeling overwhelmed by one’s problems”. The reliability coefficient Cronbach’s alpha, calculated for this study, was determined to be 0.87.

#### 2.2.2. Self-Stigma

We employed the Self-Stigma Scale-Short Form (SSS-S) by Mak and Cheung [46] to assess the self-stigma experienced by divorced single-parent women. The scale consists of nine questions, and each question is rated on a 5-point scale, ranging from strongly disagreeing (1 point) to very strongly agreeing (5 points). A higher the score on this scale indicates a greater level of generalized self-devaluation and internalization of the perceived stigma related to the individual’s single-parent status. Specific examples of the questions include statements like “The fact that I am a parent of a single-parent family makes it difficult for me”. “The fact that I am a parent of a single-parent family ruins my life”, and “There seems to be nothing I can do as a parent of a single-parent family”. The reliability coefficient Cronbach’s alpha for this study was found to be 0.89.

#### 2.2.3. Self-Esteem

Rosenberg’s [47,48] self-esteem scale was utilized to evaluate the self-esteem of divorced single-parent women. On this scale, self-esteem is defined as the level of self-respect and the extent to which one views oneself as a valuable individual [49]. The scale comprises a five-point Likert scale, with response options ranging from “Strongly disagree” (1) to “Strongly agree” (5). To interpret high self-esteem, negative questions (Nos. 3, 6, 8, 9, and 10) were reverse-scored. The calculated value of the internal consistency coefficient Cronbach’s alpha for self-esteem in this study was 0.89.

#### 2.2.4. Control Variables

In this study, various demographic background variables that are believed to influence the mental health of divorced single-parent women were considered as control variables. These variables include age, level of education, monthly income, period of single-parenthood, work status, childcare payment status, social support status, and physical health status. Specific codes were assigned to the variables as outlined below, and these codes were subsequently employed in the analysis: (1) Age (1 = “20 s”, 2 = “30 s”, 3 = “40 s”, 4 = “50 s”, 5 = “over 60 s”); (2) Level of education (1 = “Elementary school”, 2 = “Middle school”, 3 = “High school”, 4 = “Two-year college”, 5 = “Four-year university”, 6 = “Graduate or higher”); (3) Monthly income (1 = “No income”, 2 = “Less than one million won”, 3 = “One million won–two million won”, 4 = “Two million won–three million won”, 5 = “Three million won–four million won”, 6 = “Four million won–five million won”, 7 = “More than five million won”); (4) Period of single-parenthood (1 = “Less than 3 years”, 2 = “More than 3 to less than 5 years”, 3 = “5 to less than 10 years”, 4 = “More than 10 years”); (5) Work status (0 = “Currently not working”, 1 = “Currently working”); (6) Childcare payment status (0 = “Had never received”, 1 = “Have received more than once”); (7) Social support status (0 = “None”, 1 = “Have social support”); (8) Physical health status (1 = “Very poor”, 2 = “Poor”, 3 = “Fair”, 4 = “Good”, 5 = “Very good”).

### 2.3. Data Analysis

For this study, data analysis was conducted using the SPSS 25.0 (IBM Co., Armonk, NY, USA) and PROCESS Macro Version 4.1 (Model 4) [49] programs. Firstly, descriptive statistical analysis and frequency analysis were performed to examine the participants’ demographic characteristics and the main variables, including measures like mean, standard deviation, percentage, skewness, kurtosis, and range. Secondly, Cronbach’s alpha was calculated to assess the reliability of the employed scale, and correlation analysis was conducted on the main variables to examine their relationships. Thirdly, Hayes’s Process Macro Model 4 was applied to investigate the impact of self-esteem on the relationship between self-stigma and the mental health of divorced single-parent women. To verify the indirect effect, bootstrapping with 5000 resamples was used, and the mediating effect was determined by examining whether 95% confidence interval included the value of 0. A significant mediating effect was interpreted when the value of 0 was not within the confidence interval. Mean-centering was employed to address multicollinearity issues among independent variables [50], as illustrated in Figure 1 below.

## 3. Results

### 3.1. Descriptive Statistics and Correlation Analysis of Variables

Table 2 presents the results of the descriptive statistics and correlation analysis for the variables examined in this study. Firstly, the descriptive statistical outcomes (mean, standard deviation, and range) of the variables are as follows: The average score for mental health was 2.98, which indicates a level close to the average mental health status. The average score for self-stigma was 2.69, slightly below the scale’s median score of 3 points. The average self-esteem score was 3.08, indicating a normal level of self-esteem. Before examining the mediating effect, Pearson’s correlation coefficient was used to determine the relationship between the variables. The results indicated that self-stigma had a positive correlation with mental health (r = 0.53, *p* < 0.01), while self-esteem had a negative correlation with mental health (r = −0.74, *p* < 0.01). The skewness and kurtosis values met the criteria for a normal distribution with absolute values within 3 for skewness and within 8 or 10 for kurtosis [51].

### 3.2. The Mediating Effect of Self-Esteem on the Effect of Self-Stigma on Mental Health

Table 3 and Figure 2 present the findings from the analysis, examining the mediating effect of self-esteem on the relationship between self-stigma and the mental health of divorced single-parent women, while considering control variables. Firstly, it was observed that women experiencing higher levels of self-stigma tended to have lower self-esteem (B = −0.37, *p* < 0.001). Secondly, there was a negative association between self-esteem and complaints about mental health distress (B = −0.46, *p* < 0.001). Thirdly, a positive relationship was found between the level of self-stigma and the extent of complaints about mental health distress (B = 0.11, *p* < 0.001). As a result, self-esteem was identified as a partial mediator in the relationship between self-stigma and the mental health of divorced single-parent women.

Table 4 displays the outcomes of utilizing bootstrapping to assess the statistical significance of the mediating effect of self-esteem on the mental health of divorced single-parent women. The mediating effect was confirmed to be significant, as the confidence interval for the influence of self-esteem on mental health did not include 0 (B = 0.17, 95% CI = 0.13~0.22)

## 4. Discussion

This study aimed to explore the role of self-esteem as a mediator between self-stigma and the mental health of divorced, single-parent women. The results of the mediating analysis showed that self-esteem partially mediated the relationship between self-stigma and mental health. In other words, when divorced single-parent women experienced higher levels of self-stigma their self-esteem declined, leading to greater mental health distress. These findings align with the process described by Corrigan and colleagues, which suggests that internalized self-stigma can result in harmful consequences when individuals recognize and internalize negative public stereotypes related to their situation [17,18]. Moreover, the results are consistent with previous research indicating that divorced women or women from single-parent families who internalize public stigma often face low self-esteem and psychological difficulties [9,20]. In essence, this study quantitatively demonstrated the relationship between psychological challenges, stigma, and inadequate personal internal resources in divorced single-parent families, which has been qualitatively identified in previous research It highlights that the vulnerabilities faced by divorced single-parent families, which have been a subject of study for many years, still persist in contemporary times.

Despite considering demographic factors, such as level of education, monthly income, employment status, child support payment, and social support, self-stigma and self-esteem were found to significantly influence the mental health status of divorced single-parent women. These results highlight the significant role of individual internal factors in the mental health of divorced single-parent women. This observation is in line with prior research [5,29,30,32] indicating the stable and positive function of high self-esteem and the crucial nature of internalizing public stigma for divorced single-parent women. This supports the argument of Vogel et al. [52] that, even in the presence of social stigma, individuals can manage its impact at a personal level. In other words, divorced and single-parent women may experience less mental health distress if they assert their unique life values as individuals, rather than allowing societal norms, media, and welfare policies to dictate their self-perceptions. Furthermore, Reitz [53] proposed that responses and interpretations of atypical and negative life events, such as divorce and single parenting, vary among individuals, leading to diverse outcomes. Therefore, the results of this study can serve as a foundation for recognizing individual differences in response to the same life event. In essence, the subjective perceptions of divorced single-parent women, such as self-stigma and self-esteem, appear to play a more significant role in their mental health than the divorce and single-parenting events themselves.

Furthermore, the findings of this study underscore the sense of control and agency possessed by divorced single-parent women. Certain women from single-parent families exhibit a proactive attitude, wherein they maintain self-belief and confidence by taking charge of shaping their own identities [9]. In the context of divorce, individuals may view it as a voluntary means of freeing themselves from an unhappy marriage, essentially deciding against remaining in an unsatisfactory union. This perception of exercising willpower and control, coupled with the establishment of a self-image as a responsible mother among single parents, can be seen as a strategy to mitigate the risk of self-stigmatization for divorced single mothers. Divorced single-parent women can potentially avoid self-stigmatization by embracing their freedom from unhappy marriages through divorce, recognizing their autonomy in making voluntary choices, and having faith in their ability to be good mothers even in the role of single parents. A study by Leonard and Kelly [8] revealed that presenting single-parent women as excellent mothers became a way to overcome stigma and shame. Moreover, as explained by Stets and Burke [54], developing a sense of identity serves as the foundation for self-esteem, achieved by integrating social or group identity with self-worth, identity and role self-efficacy, personal identity, and self-truthfulness and authenticity. From this standpoint, it becomes evident that divorced single-parent women can foster high self-esteem by identifying themselves as valuable members of society, competent mothers, and individuals who act in alignment with their beliefs and values. In other words, despite the potential risk of self-stigmatization, it is plausible that divorced single-parent women can safeguard themselves from negative consequences, such as low self-esteem, self-stigma and shame, by independently employing strategies to protect their self-esteem.

Conversely, it is essential to implement a policy that safeguards the self-esteem of divorced single-parent women to alleviate psychological challenges. In the situation of single parents, child rearing is more likely to be a parent’s sole responsibility, and as mentioned above, depending on subjective perception, burdens and concerns about performing a complete parental role can be formed internally, so mental health difficulties can be experienced. Therefore, child rearing plays a crucial role in influencing the self-esteem of divorced single mothers. Family-friendly policies and advancements in gender equality can help mitigate the disadvantages faced by single-parent women [55]. Research by Sodermans and colleagues [56] indicated that divorced mothers with joint custody experienced positive effects by dedicating more time to leisure activities compared to those with sole custody. Bauseman [57] also found that divorced fathers with shared custody demonstrated higher levels of self-esteem. Furthermore, Biblarz and Stacey [58] revealed that families with at least two committed and compatible parents were generally more beneficial for their children, regardless of the parents’ gender or marital status. As a result, policies that reduce the burden of single-handedly raising children and encourage a shared parental role can prove effective in enhancing the overall well-being of divorced single parents and their children.

Additionally, it is obvious that the stigma that divorced single-parent women experience will take many different forms as a result of cultural differences at both the individual and societal levels. In a study conducted by Stavrova and Fetchenhauser [59], it was found that collectivist societies with strong norms favoring two-parent families had negative implications for single parents, leading to lower life satisfaction and the experience of more negative emotions. Pollmann-Schult [55] further elucidated that women’s family policies and gender equality levels varied across different regions. However, research on the effectiveness of self-stigma is primarily limited to the Western contexts and is focused on mentally ill individuals and adolescents in crisis. In Asian countries, most studies on stigma experienced by divorced single-parent families predominantly concentrate on social stigma. This likely reflects the trend that stigmatization of groups facing prejudice is more pronounced in Asian countries compared to the Western context. Future research should prioritize examining the effects of self-stigma while considering the social and personal ramifications of stigma experienced by divorced single-parent families, taking cultural specificities into account.

To sum up, divorced single-parent women are likely to encounter challenges related to their personal and social respect, as well as mental health issues. Nonetheless, these difficulties can be managed at an individual level. Hence, it is crucial to create a social and institutional environment that fosters their self-perception as valuable and positive individuals, enabling them to maintain high levels of self-esteem and live independently. Additionally, as proposed by Corrigan and Rao [17], investigating the prerequisites for negative outcomes is essential to comprehend the impact of stigma. They suggested that the effect of self-stigma on self-esteem may be partial, indicating that other factors may also contribute to negative outcomes, possibly influenced by low self-esteem. In this study, control variables such as level of education, monthly income, and work status were considered in relation to the mental health of divorced single-parent women; however, subsequent research indicates that these factors should be addressed collectively and in conjunction with other relevant factors.

## 5. Limitations

First, concerning the selection of participants, greater attention should be given to thoughtful and meticulous interpretations by referring to the reason for divorce, because in some cases this may be assessed by a woman as liberation from a crisis situation and is associated with higher self-esteem. In addition, divorced single-parent women were studied through random sampling, but there is a limit to their generalizability, necessitating follow-up studies that examine more subjects. Second, the period of being a single parent/mother was not considered, even though it can impact self-esteem. When considering that self-esteem would be sensitive to the time elapsed since the divorce, it was possible to check the level of self-esteem of women up to 3 years after divorce, between 3–5, between 5–10 and more than 10 years in this present study. A follow-up study needs to consider this issue for women’s adaptation after divorce. In this manner, the study encompassed as many control variables as possible to investigate the relationship between main variables. However, future studies should consider delving deeper into this relationship by further subdividing the control variables. Lastly, it is important to note that this study solely focused on divorced single-parent women, but single-parent males also encounter challenges as household heads, emphasizing the necessity for subsequent research to explore their experiences and circumstances.

## 6. Conclusions

This study aimed to explore the mediating role of self-esteem in the relationship between self-stigma and mental health among 347 divorced single-parent women. The results showed that self-esteem does indeed mediate the association between self-stigma and mental health, wherein higher levels of self-stigma are linked to poorer mental health outcomes. Moreover, the study highlighted that, as divorced single-parent women internalize self-stigma, their self-esteem diminishes, leading to increased mental health challenges. The study’s implications are threefold. Firstly, it emphasizes the practical significance of considering variables related to the mental health of divorced single-parent women, beyond just “perseverance”, while taking parenting and economic factors into account as control variables. Secondly, it underscores the importance of considering self-esteem over the entire lifespan of these individuals, given its relevance across different life stages. Lastly, the findings emphasize the critical role of internal variables, such as self-stigma and self-esteem, in shaping the well-being of divorced single-parent women. This insight is essential in guiding the development of policies aimed at supporting and assisting these women in their unique circumstances. 

## Figures and Tables

**Figure 1 behavsci-13-00744-f001:**
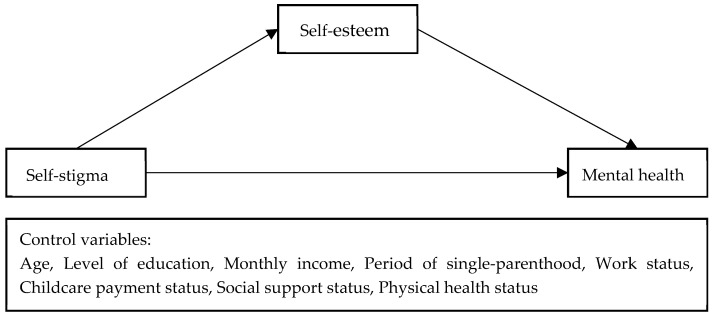
Research model.

**Figure 2 behavsci-13-00744-f002:**
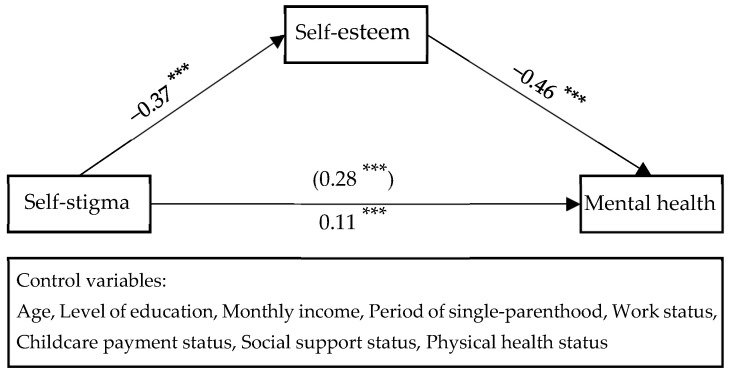
The mediating effect of self-esteem on the effect of self-stigma on mental health *** *p* < 0.001.

**Table 1 behavsci-13-00744-t001:** Demographic and sociological characteristics of participants (N = 347).

Variable		N (%)
Age	20 s	6 (1.75%)
	30 s	76 (21.9%)
	40 s	202 (58.2%)
	50 s	62 (17.9%)
	Over 60 s	1 (0.3%)
Level of education	Elementary school	4 (1.2%)
	Middle school	21 (6.1%)
	High school	181 (52.2%)
	Two year college	76 (21.9%)
	Four year university	63 (18.2%)
	Graduate or higher	2 (0.6%)
Monthly income (KRW) *	No income	20 (5.8%)
	less than one million won	92 (26.5%)
	one million won~two million won	208 (59.9%)
	two million won~three million won	22 (6.3%)
	three million won~four million won	1 (0.3%)
	four million won~five million won	2 (0.6%)
	more five million won	2 (0.6%)
Period of single parenthood	less than 3 years	102 (29.4%)
	more than 3 to less than 5 years	60 (17.3%)
	5 to less than 10 years	115 (33.1%)
	more than 10 years	70 (20.2%)
Work status	Currently working	248 (71.5%)
	Currently not working	99 (28.5%)
Child support payment status	Have received more than once	171 (49.3%)
	Had never received	176 (50.7%)
Social support status	Have social support	228 (65.7%)
	None	119 (34.3%)
Physical health status	Very poor	51 (14.7%)
	Poor	90 (25.9%)
	Fair	156 (45.0%)
	Good	42 (12.1%)
	Very good	8 (2.3%)

* KRW refers to the won, the Korean currency. Using the exchange rate in effect at the time of data collection, KRW 1159 was equivalent to roughly $1 USD.

**Table 2 behavsci-13-00744-t002:** Results of descriptive statistics and correlation analysis.

	1. Mental Health	2. Self-Stigma	3. Self-Esteem
1. Mental health	1		
2. Self-stigma	0.53 **	1	
3. Self-esteem	−0.74 **	−0.54 **	1
Mean (M)	2.98	2.69	3.08
Standard Deviation (SD)	0.54	0.93	0.67
Range	1–4	1–5	1–5
Skewness	0.10	0.16	−0.28
Kurtosis	−0.03	0.13	0.22

** *p* < 0.01.

**Table 3 behavsci-13-00744-t003:** The mediating effect of self-esteem on the effect of self-stigma on mental health.

Variable		Dependent Variable					
		Mental Health		Self-Esteem		Mental Health	
		B(SE)	95% CI	B(SE)	95% CI	B(SE)	95% CI
CV	Age	−0.11 ** (0.03)	[−0.18, −0.04]	0.17 *** (0.04)	[0.08, 0.26]	−0.03 (0.03)	[−0.09, 0.02]
	Level of education	−0.03 (0.02)	[−0.08, 0.02]	0.09 ** (0.03)	[0.03, 0.15]	0.01 (0.02)	[−0.03, 0.06]
	Monthly income	−0.01 (0.03)	[−0.07, 0.06]	−0.06 (0.04)	[−0.14, 0.02]	0.01 (0.03)	[−0.08, 0.02]
	Period of single-parenthood	−0.01 (0.02)	[−0.05, 0.03]	0.01 (0.03)	[−0.04, 0.07]	−0.01 (0.02)	[−0.04, 0.03]
	Work status	0.04 (0.06)	[−0.07, 0.15]	0.03 (0.07)	[−0.11, 0.17]	0.06 (0.05)	[−0.03, 0.15]
	Childcare payment status	0.00 (0.05)	[−0.09, 0.09]	−0.05 (0.06)	[−0.16, 0.07]	−0.02 (0.04)	[−0.09, 0.05]
	Social support status	−0.10 (0.05)	[−0.19, −0.00]	0.04 (0.06)	[−0.09, 0.16]	−0.08 (0.04)	[−0.16, −0.00]
	Physical health status	−0.21 *** (0.03)	[−0.26, −0.16]	0.17 *** (0.03)	[0.11, 0.24]	−0.13 *** (0.02)	[−0.17, −0.08]
IV	Self-stigma	0.28 *** (0.02)	[0.23, 0.32]	−0.37 *** (0.03)	[−0.42, −0.30]	0.11 *** (0.02)	[0.06, 0.15]
MV	Self-esteem					−0.46 *** (0.03)	[−0.53, −0.39]
	F	28.74 *** (0.43)		23.78 *** (0.39)		57.38 *** (0.63)	

Note. CV (Control Variable); IV (Independent Variable); MV (Mediating Variable). ** *p* < 0.01; *** *p* < 0.001.

**Table 4 behavsci-13-00744-t004:** Bootstrapping outcome of indirect effect of self-esteem on mental health.

Path	B	Boot SE	95% CI	
			Boot LLCI	Boot ULCI
Self-stigma→Self-esteem→Mental health	0.17	0.02	0.13	0.21

## Data Availability

Not applicable. The participants in this study did not provide written consent for their data to be shared publicly; therefore, due to the sensitive nature of the research, the data is not available.

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
