# Peer review of "Self-Stigma and Mental Health in Divorced Single-Parent Women: Mediating Effect of Self-Esteem"

_behavsci, 2023, doi:10.3390/bs13090744_

Round 1
Reviewer 1 Report
Abstract: Research design, sampling technique, names of the tools used, statistical techniques used in the study, other details about the sample are not mentioned in the abstract.
1.4 Present study: The first two are confusing as if is mentioned as the study is done among single parent families. The whole section itself is not clear.
The variable referred as 'self-impression' instead of self-stigma in the research question mentioned at the end of section 1.4. It is not correct.
2.1 Participants and Procedure: The first sentence states that “This study conducted one-on-one interviews or online surveys with a random sample of single-parent families residing in Ulsan, Republic of Korea”. This is confusing.
And no mention of how the single parent women were chosen. At, the end it is mentioned as randomly chosen not how and the details of the smiling technique is a major missing
Sampling technique, Research design and anything related to ethical consideration (eg: Informed consent) are not mentioned in the participants and procedure section.
2.2.1: The mental health measure used by the researcher in this study is General health Questionnaire(GHQ). In this section it mentioned as mental health scale. I think that is not appropriate.
2.2.4 Controlled variable: How the controlling was taken care of?
3.1 Is the term ‘Mental Health Pain’ appropriate?
Table 2: There is a spelling mistake in the term ‘Self-esteem’
4. Discussion: The discussion provides a comprehensive interpretation of the results in light of existing literature.
6. Conclusion: First sentence is something that we should not add in a manuscript.
(Implication mentioned in the last part of conclusion. It can be added as a separate section.)
Style of writing is not very clear
Grammatical and spelling mistakes should be avoided
Reviewer 2 Report
Thank you for the opportunity to review the manuscript. Overall, a current topic for a broader readership and further exploration of this topic is certainly unique, especially to examine the mediating effect of self-esteem in the relationship between self-stigma and mental health in
divorced, single-parent women in Korea.
A few questions / comments and suggestions:
In Line 161, stated ‘this study conducted one-on-one interviews or online surveys’, please explain clearly for your data collection mode of this study, relevant to the study is not clear.
In your discussion, in Line 355-358, explain detail for the voluntary choice of liberation, relevant to the study is not clear.
In Line 370, what meaning of negative outcomes, relevant to the study is not clear.
In Line 373, more elaboration of psychological difficulties, relevant to the study is not clear.
In Line 385, clearly present what are ‘social and intimate ways’, relevant to the study is not clear.
In Line 315-317, clearly present which different studies, relevant to the study is not clear.
In Line 387, what is ‘a negative impact on the lives of single parents’, relevant to the study is not clear.
In Line 451-453, mentioned ‘ethical review and approval were waived for this study’, is it only this study have no ethical approval for research study or all research studies in Korea?
Minor editing of English language required.
Reviewer 3 Report
I congratulate the authors of their work on undertaking a scientifically interesting and socially important topic.
My remarks relate to the following issues, because I believe that it is important to implement the subject under consideration:
1) careful selection of the group of respondents by referring to the reason for divorce, because in some cases it may be assessed by a woman as liberation from a crisis situation and is associated with higher self-esteem. I mean controlling this variable.
2) careful selection of the group of respondents by referring to the period of being a single parent/mother, self-esteem in this case is sensitive to the time elapsed since the divorce. In the presented research, it was possible to check the level of self-esteem of women up to 3 years after divorce, between 3-5, between 5-10 and more than 10 years. The second issue relates to the basis on which these ranges were distinguished, are they significant for the course of women's adaptation after divorce? I mean controlling this variable.
3) in line 107 there is information about a negative correlation between self-esteem and mental health. However, in the further part of the paper, the results of research showing a positive correlation between these variables are presented. The higher your self-esteem, the better your mental health.
4) I did not see information about the consent of the ethics committee to conduct psychological research, it is worth supplementing the work with this information.
In conclusion, I would like to emphasize that I appreciate the advantages of getting to know the job, and I also see their huge application potential.
